# Teaching Computational Machine Learning (without Statistics)

**Katherine M. Kinnaird** [1]

## Abstract

This paper presents an undergraduate machine learning course that emphasizes algorithmic understanding and programming skills while assuming no statistical training. Emphasizing the development of good habits of mind, this course trains students to be independent machine learning practitioners through an iterative, cyclical framework for teaching concepts while adding increasing depth and nuance. Beginning with unsupervised learning, this course is sequenced as a series of machine learning ideas and concepts with specific algorithms acting as concrete examples. This paper also details course organization including evaluation practices and logistics.

## 1. Introduction

In this paper, we present an undergraduate machine learning course taught at Smith College that focuses on building the habits of mind one needs to be an effective machine learning practitioner. Instead of being a survey course that "covers" as many algorithms as possible, this course seeks to leverage specific machine learning algorithms as concrete examples of larger themes and ideas in machine learning through a computational lens.

In many ways, the course presented in this paper varies from traditional approaches to teaching machine learning. First and foremost, the course begins with unsupervised learning before turning to supervised learning and then ending with a unit on deep learning. Second is the course's dual emphasis on students developing practical skills as well as deep conceptual understanding. Finally, this course places a premium on understanding machine learning concepts through heuristics and implementing pseudocode over learning complex mathematical and statistical derivations.

[1]Statistical & Data Sciences Program and Department of Computer Science, Smith College, Northampton, Massachusetts, USA. Correspondence to: Katherine M. Kinnaird <kkinnaird@smith.edu>.

*Teaching Machine Learning Workshop at ECML-PKDD 2020*, Virtual (formerly Ghent, Belgium) Sept. 14, 2020. Copyright 2020 by the author.

This paper is organized into two main parts. Section 2 motivates this computational lens for teaching machine learning. This section also includes discussions of both the expected audience and the course learning objectives. In Section 3, we delve into several course design choices, including the framework for introducing machine learning concepts, the list of topics, and general course organization from student evaluation to daily logistics.

## 2. Motivation for the course

Previous iterations of a machine learning course at Smith College were cross-listed between the Department of Computer Science department and Program in Statistical & Data Sciences (SDS). As part of an effort to broaden the appeal of a machine learning course, previous iterations had no required prerequisites, meaning that students with no training in computer science, statistics, or mathematics could take a machine learning course.

While this approach certainly can have merits, ultimately the computer science department and SDS program decided to create two courses that would complement each other and would provide discipline specific lenses for the subject matter. The SDS program offers "Modeling for Machine Learning" which emphasizes the modeling perspective for machine learning, highlighting more of the theory underpinning machine learning concepts. The computer science department offers "Computational Machine Learning" which emphasizes algorithmic understanding and programming skills. This second course is presented in this paper.

### 2.1. Key Questions & Learning Objectives

This course has a number of motivations. The beginning of the course syllabus[1] details three motivating questions posed to students: 1) What is Machine Learning? 2) What role does computer science play in machine learning? 3) What habits of mind do we need to develop to become machine learning practitioners?

Five course learning outcomes follow these motivating questions. These learning objectives are specified both to ground the course topics and activities but also to give students an

[1]Fall 2019 Syllabus can be found at http://bit.ly/CompMLsyllabus

understanding of how we will approach answering these motivating questions. As stated on the syllabus: "[b]y the end of the course, students will be able to...

- Detail differences between supervised and unsupervised learning tasks and methods, as well as discuss the issues when dealing with large scale data

- Implement a variety of machine learning algorithms in python and assess their efficacy

- Compare models, and assess the efficacy of machine learning algorithms and results using evaluation metrics and in terms of the context of the data's domain

- Develop an appreciation for ethical implications of machine learning algorithms

- Work collaboratively and reflectively to apply machine learning techniques to a data set of interest with informative documentation, written for a variety of audiences"

Both the motivating questions and the course learning objectives place emphasis on *critical thinking* in the context of machine learning. In other words, instead of focusing on memorizing a number of algorithms, formulas, and derivations, this course seeks to create machine learning practitioners with a strong foundation for conducting independent work in machine learning. Students leave this course with the ability to reason in prose and in code, to work collaboratively and iteratively, and to evaluate machine learning concepts and algorithms. This emphasis on critical thinking within a discipline specific context is inline with the liberal arts education model, common in the United States.

### 2.2. Target Audience & Expected Skills

This course was offered in a computer science department at Smith College, a small liberal arts college (SLAC) and a historically women's college. The college has about 2600 undergraduate students[2] and observes a semester system with two 15 week terms. This year, 56 students graduated with degrees in computer science. This past year, students in this course had taken fewer than 12 courses in computer science, and all self-identified as women.

This course was designed for undergraduate computer science students in their third year. Students are expected to have proficiency in programming, fluency with coding concepts, as well as ideas from theoretical computer science; and so, the computer science prerequisites for the course include introductory programming (in Python), data structures (in Java), and theory of computation (or algorithms).

Even though the emphasis of this course is on the programming aspects of machine learning, students are required to have a course in either linear algebra or multi-variable calculus.[3] This mathematical prerequisite means that while the instructor can not assume linear algebra nor multi-variable calculus, they can assume a level of mathematical maturity. This course has no statistics prerequisite.

Finally, this course makes few assumptions about students existing auxiliary programming skills. For example, with introductory programming as the only required Python course, students are not expected to have familiarity with `numpy`, `scipy`, `matplotlib`, nor `pandas`. Additionally, students are not expected to be agile with `git`, unit testing, nor with continuous integration. All of these auxiliary skills are woven throughout this computational machine learning course, adding layer of practicality to the class. In many ways, this course can be viewed as an advanced Python course motivated by machine learning.

## 3. Course Design

This machine learning course aims to both practical and conceptual. The course design focused on creating and maintaining habits of effective machine learning practitioners. Cyclical iteration is a central theme in terms of developing programming skills and for honing critical thinking.

In this section, we will introduce the framework for motivating, introducing, deploying, and practicing each machine learning concept. We will then demonstrate how the sequence of topics follows a broader path of cyclic iteration, forming a sort of cognitive outer loop to the inner loop introducing each topic. Lastly, we will illustrate how students iteratively gain professional skills throughout the semester's activities, homework assignments, and projects.

### 3.1. Cyclic Framework for Introducing Concepts

Developing critical thinking requires both repetition and reflection. For this course, each topic is first introduced through the associated heuristics. In that introduction, specific programming issues are highlighted, often through the think-pair-share discussion model (Mazur, 1997) (which is also used in other steps). The next step is crafting pseudocode that connects to the discussed heuristics. Then students create code that matches their pseudocode. In this step, students will encounter programming nuances and edge cases that need to be addressed. Finally, students are introduced to "off the shelf" implementations (perhaps from `scipy` and `sk-learn`) of the associated concept. In this final step, students may be asked to investigate the source

---

[2]Though there are post-baccalaurate programs at Smith College, there is no such program in the computer science department.

[3]At Smith College, only the second semester of calculus is a prerequisite for linear algebra, meaning that students can take linear algebra before taking multi-variable calculus.

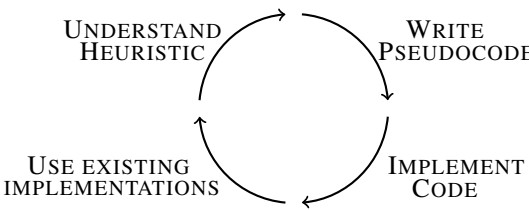

*Figure 1.* Cyclical framework for introducing concepts

code and compare it to their implementations. This cycle is illustrated in Figure 1.

In this cyclic process, students work with each concept four times. Each phase builds on the previous one, bringing new depth and nuance to student's understanding of each machine learning topic. By starting with heuristics and pseudocode as distinct phases in the learning process, the course models that explaining technical content to a broad audience is just as valuable as coding difficult algorithms. Furthermore, by continuously using this iterative framework, students have a model for expanding their machine learning toolbox after this course ends.

By engaging with each concept reflectively, students hone their critical evaluation skills, noting limitations of each machine learning approach and being able to compare different machine learning techniques. This makes transitioning from one concept to the next as straightforward as recalling limits in previous approaches or adding one new idea. For example, after students have facility with k-means, motivating kNN can be as simple as asking "what would change in your algorithm if you knew which groups 10 of your data points belong to?" With just one question, the next iteration of the cycle begins but on a new concept.

### 3.2. Topics

With this cyclical approach, it is critical to design the course with a careful scaffolding of the machine learning ideas. In this view, instead of tethering the course schedule to a list of algorithms, we assemble a sequence of themes that build on each other, pulling motivation from previous topics. For example, leveraging students' familiarity with linear regression, one can demonstrate the necessity the using both training and testing errors to avoid overfitting. It then becomes natural to ask about how to choose the "right" values for parameters and hyperparameters, which motivates introducing grid search, gradient descent, and cross-validation. Table 1 presents the ordering of the topics as both the broader machine learning ideas and the motivating examples.

While many topics are omitted from this course, the big themes of machine learning (such as comparing supervised and unsupervised learning, the train/test paradigm, etc) are present. With this course's emphasis on the programming

*Table 1.* Topics in "Computational Machine Learning" connected to the specific algorithms serving as examples

| Concepts & Themes | Examples |
|---|---|
| Auxiliary Programming Skills | `GIT`, Unit Tests, `MATPLOTLIB` |
| Distance Computations and Stopping Conditions | K-Means |
| Dimension Reduction | PCA, SVD |
| Introduction to Supervision | KNN |
| Train/Test Paradigm | Linear Regression |
| Parameters & Hyperparameters | Comparison Between KNN, K-Means, and Linear Regression |
| Tuning Parameters | Grid Search, Gradient Descent, Cross-Validation |
| Classification | SVM |
| Interpretability | Decision Trees |
| Ensemble Methods | Random Forests |
| Design Considerations | Epochs, Benchmarking |
| Deep Learning | Perceptrons, MLP, Backpropagation, `TENSORFLOW` |

aspects of machine learning, included algorithms leverage existing coding skills to illuminate machine learning ideas. What is more, this course presents a sequence of topics that do not rely on deep statistical or mathematical training.

The course adds concepts one at a time building from unsupervised machine learning to supervised and on through to deep learning, using specific algorithms as motivation for those concepts. The course's ordering of machine learning topics is less traditional but allows for a coherence across the semester with new machine learning ideas added methodically bit by bit. This approach contrasts the typical course that starts with supervised learning, then 'removes' supervision before returning to deep learning. Additionally by starting with unsupervised learning, this course highlights unsupervised learning as a critical area of study, instead of as a concession when compared to supervised learning.

This sequence of topics is not typical for a machine learning course, which makes selecting one book (or even two books) for the course challenging. Instead of drawing course readings from one book, this course drew on a number of sources including two books (Raschka & Mirjalili, 2017; McKinney, 2017), and a variety carefully curated blog and medium posts that do not rely on familiarity with supervised learning in their explanations of unsupervised learning.

### 3.3. Course Organization

This course is structured to be an active classroom with students engaged through discussion, writing, and coding nearly every class meeting. The organization of the course falls into two categories: 1) evaluation structure and 2) logistics.

#### 3.3.1. EVALUATION

The work in this course has an explicit scaffolding, leading students towards independent machine learning work. The labs are highly-structured, yet interactive. The homework emphasizes practicing implementing concepts from end to end without the step-by-step structuring of the labs. The projects are open-ended and designed to stretch students' understanding of concepts, by having students experiment and mix several machine learning ideas. These project as well as the final portfolio emphasize applying course concepts. Student work in this course falls into four types:

1. Routine course work including 26 lab assignments, weekly reflection forms, and one summary of an academic paper on fairness in machine learning

2. Coding assignments, including seven homework assignments and five projects

3. Final portfolio of revised work from the semester

4. Professional development, including two reflective writing assignments and class engagement in person and on the course's slack workspace[4]

This course uses a mix of grading practices. Homework assignments, projects, and the final portfolio are graded with typical points-based rubrics. The lab assignments, weekly reflection forms, and reflective writing assignments are graded on completion. Lastly, class engagement is graded with student input from two self-evaluation forms: one assigned in the middle of the term, and the second assigned at the end. Additionally, the course uses a series of flexibility systems designed to be more closely aligned with the "real world." For example, late work is not accepted, but to get full credit for several areas of the course, students need to do about 80% of the work well. This model allows students to practice making decisions such as balancing health and work in a more authentic manner than under a traditional points-based grading where every point must be attempted.

#### 3.3.2. COURSE LOGISTICS

The course logistics have two goals: 1) students engage actively with concepts and 2) students practice skills expected

by industry. Explicitly the course was designed such that students repeatedly engaged with version control, continuous integration, and weaving code with prose.

Instead of using a typical learning management system like Canvas or Blackboard, this course uses a combination of GitHub Classroom and Travis CI for managing course materials and collecting assignments. Course set-up in GitHub Classroom followed recommendations from Fiksel et al. (2019).[5] Additionally homework assignments and projects include unit testing and continuous integration through Travis CI.[6] This set-up required continuous practice of auxiliary programming skills. Given that `git`, GitHub, and continuous integration are not widely adopted in the computer science department at Smith College, students set up these services through the first lab, which was adapted from the first three parts of McFee and Kell's tutorial (2018) and parts of the tutorials on GitHub Classroom associated to Fiksel et al.'s work (2019).

This course's explicit focus on practicality extends to content delivery, preferring an interactive setting to the standard lecture-based teaching model.[7] Every class meeting uses a computational notebook (either Jupyter or Colab) as the main vehicle for content delivery. Called "labs," each notebook outlines a class meeting, including short reading passages, discussion questions (with text boxes to take notes), and coding blocks for students to experiment in. Using computational notebooks allowed the instructor to structure course notes for the students and to model the iterative learning process discussed in Section 3.1. Effectively, these notebooks are machine learning worksheets, but students readily accepted them given the interactive nature inherent to Jupyter notebooks (Kluyver et al., 2016).

## 4. Conclusion

In this paper, we have presented "Computational Machine Learning," a machine learning course for undergraduate computer science majors with limited mathematical training. This course demonstrates that teaching a meaningful course in machine learning without statistical training is possible. In addition to providing a number of course logistics, we have also provided an iterative cyclical framework for introducing concepts as well as list of topics that leverage this framework to create natural transitions between topics.

---

[4]Slack - https://slack.com/ - is a kind of discussion board that can have channels for specific topics.

[5]GitHub Classroom for Teachers - https://github.com/jfiksel/github-classroom-for-teachers
Github Classroom for Students - https://github.com/jfiksel/github-classroom-for-students

[6]Travis CI - https://travis-ci.org/

[7]The initial version of course materials is available at - https://github.com/comp-machine-learning-general/course-materials.

## Acknowledgements

The author is the Clare Boothe Luce Assistant Professor of Computer Science and Statistical and Data Science at Smith College and as such, is supported by Henry Luce Foundation's Clare Boothe Luce Program. Any opinions, findings, and conclusions or recommendations expressed in this material are those of the authors and do not necessarily reflect the views of the Luce Foundation.

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
