# OpenReview forum: "Teaching Computational Machine Learning (without Statistics)"
_ECMLPKDD.org/2020/Workshop/TeachML — ECML PKDD 2020 TeachML_

### Official Review · AnonReviewer2 · 2020-07-15
**python course motivated by machine learning**

**Rating:** 7
**Confidence:** 5

**Review:**

The paper describes a course for computer science students. The course teaches concepts of machine learning with a focus on implementation of methods. On the side it teaches important skill like version control and clitical thinking.

## Pros:
The paper nicely describes this introductory machine learning course.
The focus on implementation is unusual, yet interesting.
The cyclic framework described in the paper, as well as the active classroom, the flexibility system in assignments and the use of computational notebooks and continuous integration, are exciting to see.

## Cons:
I would have hoped for a link to the teaching material or at least an example showing one of the notebooks.
Also I remain with one question: Are the teaching materials openly available? If not, why?
If space in the paper was the issue, I would rather neglect the description of the two courses (stats and CS) and the college/students .

Also, it would have been nice to read, what the students think about the coures (e.g. course evaluation results).


### Minor comments:
- Learning objectives: the first point seems like two points to me
- Learning objectives: assess efficacy is mentioned in both points 2 and 3 and thus could be deleted in point 2
- Evaluation: "mix a severeal machine learning ideas" -> "mix severeal machine learning ideas"
- Evaluation: what do you mean by "slack site"?

---

### Official Review · AnonReviewer1 · 2020-07-26
**An applied course on computational machine learning**

**Rating:** 7
**Confidence:** 5

**Review:**

The paper highlights a course on computational ML offered in a CS program. The course is intended as an accessible entry point to ML for undergraduate students, and assumes programming background and some mathematical maturity (either LA or multi-variable calculus are required).

Pros:

In addition to introducing ML and its links with CS, the course also poses the interesting question on what habits of mind are needed for ML practitioners. This is, in my experience, a necessary skill for ML practitioners where most practical problems come with unique caveats. The learning outcomes for the course likewise cover a fairly broad spectrum of topics, ranging from implementing to understanding different ML algorithms. The course also addresses ethical concerns and collaborative development, which is exciting to see at such an early stage. Use of continuous integration and unit testing is also a welcome addition to the course. The cyclic framework should be an excellent tool for reinforcing learning outcomes, I really liked the concept!

Some open questions include:
1. The authors decided to start the course with unsupervised learning algorithms rather than supervised learning. While I do not have an issue with this reordering naturally, I was wondering if the authors uncovered any evidence that one way or the other was more effective. This would be very useful for the broader community.
2. While the course covers continuous integration, is this limited to the code the students write or does it extend also to the models they train? Do students deal with concepts such as how to serialise, share and/or update their models in an online setting?
3. I would be very interested in reading some of the feedback the authors received (so tying back to point 1). Also, the author's experience of using Github Classroom and Travis CI in this setting would be very interesting. If lack of space is an issue, perhaps the authors could include this information in their presentation?

---

### Official Review · AnonReviewer3 · 2020-07-27
**Teaching students practical machine learning**

**Rating:** 8
**Confidence:** 5

**Review:**

Teaching students practical machine learning

This paper describes the structure and content of an instructional machine learning course. The focus on the practical aspects of machine learning for this course is an important idea. From personal experience with teaching Machine Learning, I can confirm that the knowledge about the mathematical background and the structure of the algorithms alone is not sufficient to apply these methods to specific problems as there are further technical challenges to solve.

**Positive**

- The focus on practical aspects of applying machine learning
- The repeated contact with concepts, in different stages of the learning process (cyclic framework)
- The idea that communicating technical concepts is an essential skill
- The usage of version control and testing is part of the course
- Bonus points for incorporating the ethical implications of ML algorithms.

**Open Questions**
- Is the focus on implementing the algorithms or applying them? For example: Implementing a k-means clustering correctly might be a different challenge than applying it to a particular problem. I am unsure whats the focus of the course.
- Some information about the content of the "model evaluation" part of the course would be great. E.g. used metrics, testing procedures etc.
- This paper gives a general overview of the topic (due to length regulations), some more details about the topics and tasks would help. It would be great if the course material would be publicly available.

**Minor Questions**

- I am unsure why Table 1. lists "Linear Regression" as "Train/Test Paradigm". Would expect something like "n-fold cross-validation".

---

### Decision · Program_Chairs · 2020-07-31

**Decision:**

Accept

**Comment:**

The reviewers agree that this paper will be accepted. Thank you for your contributions.

Please register with the conference as soon as possible! See this page for details:
https://ecmlpkdd2020.net/attending/registration/.
Which asks that at least one author per paper registers until July 31, 2020.
We apologize for the very short notice.